# Visualization and Quantification of the Extracellular Matrix in Prostate Cancer Using an Elastin Specific Molecular Probe

**DOI:** 10.3390/biology10111217

**Published:** 2021-11-22

**Authors:** Avan Kader, Julia Brangsch, Carolin Reimann, Jan O. Kaufmann, Dilyana B. Mangarova, Jana Moeckel, Lisa C. Adams, Jing Zhao, Jessica Saatz, Heike Traub, Rebecca Buchholz, Uwe Karst, Bernd Hamm, Marcus R. Makowski

**Affiliations:** 1Department of Radiology, Institute of Integrative Neuroanatomy, Charité—Universitätsmedizin Berlin, Corporate Member of Freie Universität Berlin, Humboldt-Universität zu Berlin, and Berlin Institute of Health, Charitéplatz 1, 10117 Berlin, Germany; julia.brangsch@charite.de (J.B.); carolinreimann1990@web.de (C.R.); jan-ole.kaufmann@charite.de (J.O.K.); dilyana.mangarova@charite.de (D.B.M.); jana.moeckel@charite.de (J.M.); lisa.adams@charite.de (L.C.A.); jing.zhao@charite.de (J.Z.); bernd.hamm@charite.de (B.H.); marcus.makowski@tum.de (M.R.M.); 2Department of Biology, Chemistry and Pharmacy, Institute of Biology, Freie Universität Berlin, Königin-Luise-Str. 1-3, 14195 Berlin, Germany; 3Division 1.5 Protein Analysis, Bundesanstalt für Materialforschung und-Prüfung (BAM), Richard-Willstätter-Str. 11, 12489 Berlin, Germany; 4Department of Chemistry, Humboldt-Universität zu Berlin, Brook-Taylor-Str. 2, 12489 Berlin, Germany; 5Department of Veterinary Medicine, Institute of Veterinary Pathology, Freie Universität Berlin, Robert-von-Ostertag-Str. 15, Building 12, 14163 Berlin, Germany; 6Division 1.1 Inorganic Trace Analysis, Bundesanstalt für Materialforschung und-Prüfung (BAM), Richard-Willstätter-Str. 11, 12489 Berlin, Germany; jessica.saatz@bam.de (J.S.); heike.traub@bam.de (H.T.); 7Institute of Inorganic and Analytical Chemistry, Westfälische Wilhelms-Universität Münster, 48419 Münster, Germany; rebecca.buchholz@uni-muenster.de (R.B.); uk@uni-muenster.de (U.K.); 8School of Biomedical Engineering and Imaging Sciences, King’s College London, St Thomas’ Hospital Westminster Bridge Road, London SE1 7EH, UK; 9Department of Diagnostic and Interventional Radiology, Technical University of Munich, Ismaninger Str. 22, 81675 Munich, Germany

**Keywords:** magnetic resonance imaging, molecular imaging, prostate cancer

## Abstract

**Simple Summary:**

One of the most commonly diagnosed cancers in men is prostate cancer (PCa). Understanding tumor progression can help diagnose and treat the disease at an early stage. Components of the extracellular matrix (ECM) play a key role in the development and progression of PCa. Elastin is an essential component of the ECM and constantly changes during tumor development. This article visualizes and quantifies elastin in magnetic resonance imaging (MRI) using a small molecule probe. Results were correlated with histological examinations. Using an elastin-specific molecular probe, we were able to make predictions about the cellular structure in relation to elastin and thus draw conclusions about the size of the tumor, with smaller tumors having a higher elastin content than larger tumors.

**Abstract:**

Human prostate cancer (PCa) is a type of malignancy and one of the most frequently diagnosed cancers in men. Elastin is an important component of the extracellular matrix and is involved in the structure and organization of prostate tissue. The present study examined prostate cancer in a xenograft mouse model using an elastin-specific molecular probe for magnetic resonance molecular imaging. Two different tumor sizes (500 mm^3^ and 1000 mm^3^) were compared and analyzed by MRI in vivo and histologically and analytically ex vivo. The T1-weighted sequence was used in a clinical 3-T scanner to calculate the relative contrast enhancement before and after probe administration. Our results show that the use of an elastin-specific probe enables better discrimination between tumors and surrounding healthy tissue. Furthermore, specific binding of the probe to elastin fibers was confirmed by histological examination and laser ablation–inductively coupled plasma–mass spectrometry (LA-ICP-MS). Smaller tumors showed significantly higher signal intensity (*p* > 0.001), which correlates with the higher proportion of elastin fibers in the histological evaluation than in larger tumors. A strong correlation was seen between relative enhancement (RE) and Elastica–van Gieson staining (R2 = 0.88). RE was related to inductively coupled plasma–mass spectrometry data for Gd and showed a correlation (R2 = 0.78). Thus, molecular MRI could become a novel quantitative tool for the early evaluation and detection of PCa.

## 1. Introduction

Prostate cancer (PCa) accounts for one in five cancer diagnoses in men, making it one of the most commonly diagnosed carcinomas in men in the Western world [1]. It can be a highly malignant tumor disease, and represents one of the most common fatal cancers in men [1]. The causes of the disease are yet not fully understood. Risk factors include age, ethnic origin, geographical location, and genetic predisposition [2,3,4,5]. In the last few years, both the diagnosis and treatment of PCa have improved due to medical advances.

Early-stage prostate-specific antigen (PSA) screening is considered practical for decision-making and treatment in PCa [2]. The laboratory-chemical survey determination of the PSA level is well established in Western countries, but this method also shows significant limitations. Due to its relatively low specificity and a low sensitivity, it leads to many false positive diagnoses [6,7]. As a result, many patients undergo unnecessary prostate biopsy [7]. In patients with a normal PSA level, PCa could be diagnosed in 30% of cases, with 10% being assigned to aggressive PCa [6]. The PSA value therefore does not necessarily provide any information about the aggressiveness of the tumor. Other factors can also affect PSA level, such as bacterial prostatitis and acute urinary retention [8]. PSA screening also leads to over-diagnosis of PCa and thus initiates unnecessary surgical procedures to collect tissue samples [9].

Another diagnostic option is transrectal ultrasound (TRUS) [10], which is now well established in clinical practice. TRUS can help determine the volume of the prostate and is used as a supplementary diagnostic test.

An indispensable method for the diagnosis of PCa is magnetic resonance imaging (MRI). Diffusion-weighted apparent diffusion coefficient (ADC) imaging, T2-weighted imaging, and dynamic intravenous contrast-enhanced (DCE) imaging with unspecific contrast agents are among the standard MRI examinations in PCa, referred to as multiparametric MRI (mpMRI) [11,12]. Some of the advantages of MRI are the high-resolution spatial imaging of tissue with strong soft tissue contrast, the quantitative imaging technique, and the lack of invasiveness and radiation. mpMRI has a high sensitivity and specificity, but it has a low positive predictive value (PPV) [13]. The Prostate Imaging Reporting and Data System (PI-RADS) helps to detect PCa in a standardized form. The main challenge is the different interpretation of the PI-RADS results by clinicians and medical staff [13]. PCa mpMRI diagnosis must finally be verified by biopsy. Therefore, it is necessary to develop a non-invasive screening method with high specificity.

To improve MRI contrast, the paramagnetic lanthanide metalion gadolinium (III) (Gd^3+^) can be used in a complex with organic chelates, like the macrocyclic DOTA (2,2′,2″,2′′′-(1,4,7,10-Tetraazacyclododecane-1,4,7,10-tetrayl)tetraacetic acid) or the linear DTPA (2,2′,2′′,2′′′-{[(Carboxymethyl)azanediyl]bis(ethane-2,1-diylnitrilo)}tetraacetic acid) as contrast media. Thereby, the unpaired electrons in the Gd-ion will shorten the T1 relaxivity of the neighboring water protons and thus, the signal intensity of the Ta-weighed image will be increased [14].

To further improve tissue differentiation, molecular probes could be used as opposed to the currently available unspecific extracellular Gd-based contrast agents [15]. In addition, molecular imaging could be used as an added tool to current diagnostic techniques.

A possibility to use appropriate small molecule biomarkers for the detection of malignant diseases would be to target extracellular matrix (ECM) components. These can be coupled with MRI-compatible elements. The ECM architecture plays a main role during the development and progression of PCa [16]. Palumbo et al. showed strong stimulation of the proliferation and migration of tumor cells (LNCaP) by the ECM, but also inhibition of apoptosis and deregulation of the expression of several genes [16].

The main matrix macromolecule components are elastin, collagen, fibronectin, laminin, and proteoglycan [17,18,19]. The interaction between tumor cells and elastic fibers is controlled by a 67 kDa receptor [20,21]. Although the signal mediation of the receptor in tumor cells is not yet fully elucidated, a different extensive binding of tumor cell lines to elastin has been observed in Lewis lung carcinoma cells [22]. Tumor cells are able to express, adhere, degrade, and migrate elastin proteins [23]. Lysyl oxidase, a copper-dependent aminodase, promotes the cross-linking of collagen and elastin in tissue and is responsible for the activation of the elastin promoters [24,25]. This is a determining factor in the stiffness and structural stability of ECM [24]. An interaction between the tumor cells and the ECM protein elastin is mediated by two elastin-binding proteins (S-gal and galectin-3) and two laminin receptors [23].

The expression of elastin-binding proteins is strongly related to the metastatic potential of the tumor [23]. One possible explanation is that cancer cells are able to synthesize elastin and express lysyl oxidase [23]. Calderón et al. showed that PCa contains more elastic fibers than normal tissue [26]. Elastin fibers are implicated in tumor invasion and metastasis, cell proliferation, adhesion, apoptosis, and angiogenesis [26,27,28]. Finally, elastin represents a novel promising molecular biomarker also in the field of cardiovascular diseases [29,30]. Additionally, hepatic cancer [31] could be evaluated using an elastin-specific MRI molecular probe.

Despite the advances in the diagnosis of PCa over the course of time, further studies are needed to clarify the onset and mechanism of PCa progression.

We therefore analyzed the role of elastin in conjunction with molecular MR imaging in a xenograft mouse model, comparing two different tumor sizes. This study aimed to use a low-molecular elastin-specific probe in MRI examinations and, thus, to obtain information on changes in the ECM during prostate cancer development for a better differentiation between tumor tissue and healthy tissue.

## 2. Materials and Methods

### 2.1. Cell Culture

Human PC3 cells were obtained from ATCC^®^ CRL-1435^™^ (Manassas, VA, USA) and cultured in Roswell Park Memorial Institute (RPMI) 1640 Medium (Gibco™, Thermo Fischer Scientific, Waltham, MA, USA) supplemented with 10% fetal calf serum (FCS) (Gibco™, Thermo Fischer Scientific, Waltham, MA, USA). Cells were cultured in 150 cm² tissue culture flasks until they reached about 80% confluence. Cells were washed with phosphate buffered saline (PBS) (Gibco™, Thermo Fischer Scientific, Waltham, MA, USA), trypsinized, and subsequently re-suspended in 1 mL RPMI-medium and counted with 0.4% Tryptan blue solution (Gibco™, Thermo Fischer Scientific, Waltham, MA, USA).

### 2.2. Xenograft Mouse Model

This study was performed corresponding to the local guidelines and provisions for the implementation of the Animal Welfare Act and the regulations of the Federation of Laboratory Animal Science Associations (FELASA). This animal study was approved by the regulatory authority of the Regional Office for Health and Social Affairs Berlin (LAGeSo) (G0094/19). Male, eight-week-old SCID-mice (CB17/Icr-Prkdcscid/IcrIcoCrl) were obtained from Charles River Laboratories (Sulzfeld, Germany) (N = 28). The animals were randomly assigned to two different groups (n = 14).

For anesthesia, the mice were intraperitoneally injected with medetomidin (500 µg/kg), midazolam (5 mg/kg), and fentanyl (50 µg/kg). Cell suspension with 2 × 10^6^ PC3-cells was injected subcutaneously (s.c.) in the area of the right scapula. Anesthesia was subsequently antagonized with atipamezol (750 µg/kg), flumazenil (0.5 mg/kg), and naloxon (1200 µg/kg).

MR imaging was performed on a tumor size of 500 mm^3^ (n = 14) or 1000 mm^3^ (n = 14). The size of the tumor was determined using calipers. Following MRI, mice were euthanized and tumor tissue was removed for ex vivo examination.

### 2.3. In Vivo MRI

MR imaging was performed using a 3.0 Tesla MR scanner (MAGNETOM Lumina, Siemens, Erlangen, Germany) and a 4-channel receive-coil array for mouse body applications (mouse scapula array, P-H04LE-030, Version1, Rapid Biomedical GmbH, Germany). Following s.c. anesthesia as described above, mice were positioned on the MRI patient table in a prone position. A venous access through the tail vein was established for administration of the contrast agent during the MR imaging. The body temperature (37 °C) was monitored with the use of an MR-compatible heating system (Model 1025, SA Instruments Inc, Stony Brook, NY, USA) to avoid rapid cooling.

### 2.4. Elastin-Specific Contrast Agent for the MRI

A contrast agent that specifically binds to elastin was used for the experiments (ESMA; Lantheus Medical Imaging, North Billerica, MA, USA). It is a low-molecular-weight gadolinium-based contrast agent with a molecular mass of 856 g/mol [30]. The highest binding is achieved after 30 to 45 min [30,32]. The longitudinal relaxivity of 4.68 ± 0.13 mM^−1^s^−1^ and 8.65 ± 0.42 mM^−1^s^−1^ [30,32] is known. The contrast agent was administered intravenously via the tail vein in a clinical dose of 0.2 mmol/kg.

### 2.5. Elastin Imaging Using T1 Weighted Sequences

MR imaging was realized with a 3.0 Tesla MR scanner. The mice were imaged in prone position with a 4-channel receive-coil array for mouse body applications. For the localization of the tumor, a low-resolution three-dimensional localizer scan was used, which was performed in sagittal, coronal, and transverse orientation with the following parameters: field-of-view (FOV) = 280 × 280 mm, matrix = 320, slice thickness = 1.5 mm, repetition time (TR) = 11.0 ms, echo time (TE) = 5.39 ms, flip angle = 20°, and slices = 10. Anatomic images were captured using a T2-weighted sequence with the following parameters: FOV = 150 mm, matrix = 201, slice thickness = 1.2 mm, TR = 3200.0 ms, TE = 77.0 ms, flip angle = 140°, and slices = 25. To visualize the gadolinium-based contrast agent, a T1-weighted sequence was performed with the following parameters: FOV = 70 mm, matrix = 131, slice thickness = 0.4 mm, TR = 833.8 ms, TE = 6.34 ms, flip angle = 30°, and slices = 30.

### 2.6. MRI Measurements

MR images were analyzed using Visage 7.1 (Version 7.1, Visage Imaging, Germany). The T1-weighted images were compared before and after the administration of the contrast agent (signal intensity = *SI*). For relative enhancement (*RE*) assessment, 2D regions of interest (ROIs) were drawn around the respective areas in pre-contrast and post-contrast MR images. The following formula was used to calculate the relative enhancement (*RE*):$$\mathrm{RE}=\frac{\left({\mathrm{SI}}_{postcontrast}-{\mathrm{SI}}_{precontrast}\right)}{{\mathrm{SI}}_{precontrast}}$$

### 2.7. Competition Experiment

Three mice were used for the competition experiment (n = 3). After a tumor size of 1000 mm^3^ was reached, the animals were anesthetized and examined in an MRI (MAGNETOM Lumina, Siemens, Erlangen, Germany). On day one, imaging without a contrast agent was followed by an intravenous injection of the elastin-specific contrast agent (0.2 mmol/kg). Additional MRI images were acquired as described above (*Elastin imaging using T1 weighted sequences*) and the animals were then antagonized. On day two, following a native MRI scan, a 5-fold higher dose of an elastin-specific europium-coupled contrast agent was administered through the tail vein. After this imaging, the Gd-containing elastin-specific contrast agent was administered and imaged in an MRI. The data obtained were compared for signal changes.

### 2.8. Histological Analysis

Frozen samples were cut into 9 μm-thick serial sections at −20 °C. The sections were then fixed with cold acetone (≥99%, Fisher Scientific, Hampton, VA, USA) for 6 min at −20 °C. Miller’s Elastica–van Gieson-stain (EvG) was performed. EvG was used to visualize elastic fibers. In addition, immunofluorescence staining was conducted using an anti-elastin antibody (Rabbit anti-Mouse pAb Elastin, abcam^®^, Cambridge, United Kingdom) that was diluted 1:100 with Dako REAL^TM^ Antibody Diluent (DAKO, Glostrup Denmark), and incubated overnight at 4 °C. The sections were washed three times with PBS (pH = 7.4), followed by a 1 h incubation with the secondary antibody (1:200, donkey anti-rabbit IgG, Invitrogen, Carlsbad, CA, USA). The samples were washed again three times with PBS and covered with DAPI staining solution (ROTI^®^ Mount FluorCare DAPI, Carl Roth, Karlsruhe, Germany). Last, the sections were analyzed using a Keyence microscope (BZ-x800 Series, Osaka Prefecture, Japan).

### 2.9. Quantification of the EvG Stain and Immunofluorescence

The quantification of the staining area of the EvG and immunofluorescence sections was measured with the image analysis software BZ-X800 Analyzer (Keyence, Osaka prefecture, Japan). Three representative areas (two different peripheral areas and one central region) were analyzed for each probe. The mean value was calculated in each case. First, the entire region of interest was marked. Consecutively, all elastic fibers were identified and the relation of the elastic fibers to the entire marked tumor region was calculated using marked pixels.

### 2.10. Laser Ablation–Inductively Coupled Plasma–Mass Spectroscopy (LA-ICP-MS)

LA-ICP-MS was performed for localization of gadolinium (Gd) in the tumor tissue (n = 3 per group). Tumor samples were cut into 9 µm cryosections at −20 °C and mounted on SuperFrost Plus adhesion slides (Thermo Scientific, Waltham, MA, USA).

The analysis was performed by continuously scanning the thin sections and transport of the aerosol via He-gas flow to the ICP-MS. Two different LA-ICP-MS systems were used, which are described in the Appendix A. Matrix matched gel standards were used for drift control and calibration of ^158^Gd.

### 2.11. Inductively Coupled Plasma–Mass Spectrometry (ICP-MS)

ICP-MS was used to determine total gadolinium concentrations in tumor samples. A piece of the tumor sample was prepared (n = 5 per group) and dried under a vacuum atmosphere (vacuum pumping unit, vacuubrand^®^, Wertheim, Germany). One mL of 66% nitric acid was added to each sample and incubated at room temperature until the tissue was completely dissolved. Deionized water was then added to each sample. Digested samples were diluted in 1% HNO_3_ sub-boiling (s.b.) and analyzed with an iCAP Qc ICP quadrupole mass spectrometer (Thermo Fisher Scientific, Bremen, Germany) in combination with the autosampler 4DXF-73A (ESI Elemental Service & Instruments GmbH, Mainz, Germany) using a 200 µL PFA nebulizer and a cyclonic spray chamber (see Table 1 for more details). Calibration was carried out using diluted Gadolinium ICP Standard CertiPUR (Merck KGaA, Darmstadt, Germany) and using rhodium as the internal standard. More details can be found in the Appendix A.

### 2.12. Western Blot

For protein isolation from the tissue, a tumor piece was first homogenized in RIPA buffer (n = 3 per group). For this purpose, 50 mM Tris·HCl (Carl Roth GmbH, Karlsruhe, Germany), 150 mM NaCl (Carl Roth GmbH, Karlsruhe, Germany), 0.1% SDS (Carl Roth GmbH, Karlsruhe, Germany), 1% sodium deoxycholate (Carl Roth GmbH, Karlsruhe, Germany), and 1% Triton X-100 (Merck, Darmstadt, Germany) were mixed with protease inhibitor I and protease inhibitor II (Thermo Fisher Scientific, Waltham, MA, USA). The samples were shaken shortly and shaken for 2 h at 4 °C. This was followed by centrifugation at 12,000 rpm for 20 min at 4 °C. Samples were filtered using tip filters (1 µm, 0.45 µm, 0.1 µm). The sample concentration was determined using the BC assay method (Pierce™ BCA Protein Assay Kit, Thermo Fisher Scientific, Waltham, MA, USA). The manufacturer’s protocol was used. The same protein amount (50 µg) was loaded into the wells of the gel under unreduced conditions (SERVAGel™ TG 8% PRiME™, Heidelberg, Germany) and separated in the running gel system (SERVA™ Heidelberg, Germany) at a voltage of 70 V for 60 min and then at 160 V for 60 min in running buffer (250 mM TrisBase (Carl Roth GmbH, Karlsruhe, Germany), 1.92 M glycine (Carl Roth GmbH, Karlsruhe, Germany), and 1% SDS (Carl Roth GmbH, Karlsruhe, Germany)). Subsequently, the proteins were transferred from sodium dodecyl sulphate (SDS) gel to a nitrocellulose membrane (Trans-Blot^®^ Turbo™ RTA Mini PVDF Transfer Kit, Bio-Rad Laboratories, Hercules, CA, USA). The blot system Trans-Blot^®^ Turbo™ (Bio-Rad, Laboratories, Hercules, CA, USA) was used. A 5% skim milk powder (Carl Roth GmbH, Karlsruhe, Germany) in 0.05% PBS-Tween20 (PBS-T) (Carl Roth GmbH, Karlsruhe, Germany) solution was used to block non-specific antibody binding. Incubation was performed at room temperature for 1 h. Blots were incubated with a mouse monoclonal anti-elastin antibody (sc-166543, Santa Cruz Biotechnology, Dallas, TX, USA) diluted 1:1000 in 5% milk solution overnight at 4 °C. After washing the membrane three times with PBS-T, the blots were incubated with HRP-coupled mouse IgGκlight chain binding protein diluted 1:5000 in PBS-T for 60 min. The band was detected using the membrane substrate (SeramunBlau^®^ prec, Seramun Diagnostica GmbH, Heidesee, Germany). GAPDH (Invitrogen, Carlsbad, CA, USA) was used for charge control.

The intensity of the bands was measured with the software Image J (ImageJ software, Version 1.53).

### 2.13. Statistical Analysis

A mean bet was calculated and presented from all the data. The significance was compared by unpaired and bilateral t-test analysis and significance was shown at *p* < 0.05. Statistics were performed with Microsoft Excel.

## 3. Results

In this study, a gadolinium-based elastin-specific probe was used to examine ECM changes during PCa development. Two different tumor sizes were examined. For a detailed study setup please see Figure 1.

All animals developed a tumor. Tumor growth at the same injection time was heterogenic. The final size of the tumor was determined by daily tumor measurement. The final tumor size of 1000 mm^3^ was reached after 36 to 50 days. In the other group, which developed a tumor volume of 500 mm^3^, the target volume was reached after 30 and 64 days. One animal had to be withdrawn from the trial early because of poor general condition (n = 1) (was replaced by another mouse).

### 3.1. Molecular Characterization in T1-Weighted MR Imaging Using Gd-Based Elastin-Specific Contrast Agent

The intravenous administration of the elastin-specific contrast agent resulted in a significant MR signal increase (*p* ≤ 0.001) in the area of the subcutaneous tumor in all examined mice. Figure 3B shows a pre-contrast image and Figure 3C shows an image with a contrast medium. A good difference between the two groups can already be seen here. Mice with a tumor size of 1000 mm^3^ showed a twofold increased SI, whereas mice with 500 mm^3^ tumors showed an even higher (threefold) increased SI (Figure 2A). In the group with a tumor volume of 1000 mm^3^, the SI was 3037 after contrast agent administration (pre-contrast SI of 897) (*p* ≤ 0.001). In mice with a tumor size of 500 mm^3^, after application of the elastin-specific contrast agent an SI of 3819 was determined (pre-contrast SI of 907) (*p* ≤ 0.001).

The specific binding of the contrast agent was demonstrated by a competition experiment. The administration of a europium-coupled elastin-specific probe did not provide sufficient signal enhancement, as shown in Figure 2B. These data from the previous day (pre-scan and after elastin-specific contrast agent administration) were compared with the second-day data (pre-scan, europium-coupled probe and elastin-specific contrast agent). On the first day, an SI of around 3000 was obtained after elastin-specific contrast agent administration. On the second day, the data from native imaging, the europium-coupled probe, and the elastin-specific contrast agent showed no change in RE.

To show signal enhancement within a mouse, a fusion map was created (Figure 3D) that shows the SI between the T2 and T1 sequences after contrast administration in the same mouse.

### 3.2. Detection of Elastin Fibers in Tumor Tissue with Histological Analysis

In both tumor sizes, elastin fibers were observed in the entire tumor tissue, as shown in Figure 4A. The elastin fibers were dyed blue to purple. To determine the elastin content of a sample, three areas were selected for each slide and the percentage of elastin content was determined with the analyzer. The evaluation showed a difference between the two groups: Fewer elastic fibers were detected in 1000 mm^3^ tumors compared to 500 mm^3^. The mean value of n = 14 was 3.3% (σ = 0.9) in the 500 mm^3^ and 3.0 % (σ = 0.9) in the 1000 mm^3^ tumors (n = 14) (Figure 2C). In addition, the detected values (percentage of elastin) of each tumor strongly correlated with the RE data of in vivo MRI imaging (T1-weighted MR sequences) (Figure 2D, y = 1.1304x − 0.7943, R = 0.877).

The in vivo MRI images showed an irregular distribution of the elastin fibers. This observation was also confirmed in the ex vivo histological analysis (Figure 4A).

To further evaluate the distribution of elastic fibers in the PC3 tumor, an immunofluorescence staining with an anti-elastin antibody was performed (Figure 4B). This showed an irregular distribution of the elastin fibers in the tissue. The mean value of n = 4 was 7.5% (σ = 1.8) in 500 mm^3^ and 3.7% (σ = 0.9) in 1000 mm^3^ tumors (*p* < 0.05).

The Western blot showed a lower elastin expression in the 1000 mm^3^ group compared to the 500 mm^3^ group (Figure 4D) (full WB can be found in Appendix A). For each group, n = 3 animals were evaluated. The antibody used for the Western blot was different than that used for immunofluorescence, as the respective antibodies had to be applied specifically to one method.

### 3.3. Elemental Analysis of Tumor Tissue with Specific Regard to Gadolinium

LA-ICP-MS measurements were used to localize gadolinium in PC3 tumor tissue. Three tumor sections were analyzed for each group (n = 3). Strong colocalization of gadolinium with elastic fibers was shown (Figure 4C). Here it can be seen that the peripheral area of the tumor, as well as the intra-tumoral space, contained gadolinium.

The gadolinium concentration in the tumor was quantified by ICP-MS after dissolution of the samples. The concentration of gadolinium with elastic fibers was correlated with in vivo RE data and showed a strong correlation (y = 1.7606x + 0.6126; R^2^ = 0.78; *p* ≤ 0.001) (Figure 2E). Measurements by ICP-MS were performed in n = 5 for each group.

## 4. Discussion

This study shows the feasibility of an elastin-specific MRI molecular probe for the characterization of a PC3 tumor in a SCID mouse model. The results indicate that elastic fibers have an irregular distribution across the entire PC3 tumor tissue. In all examined tumors, a high number of elastic fibers was measured, especially in the marginal area, regardless of the tumor size. Thus, a better distinction between healthy tissue and tumorous tissue was feasible. In addition, smaller tumors were found to express more elastin than larger tumors.

ECM proteins play an essential role in tumor development, cell behavior, and microenvironment. The ECM is responsible for the architecture of the tumor [33] and can change continuously [34]. The structure of the ECM in tumor diseases is essential for understanding tumor development and therefore for developing diagnostic and therapeutic options. Not only does the elasticity of a tumor depend on the ECM but also the stiffness, and it is responsible for the homeostasis of the tissue [33].

In many types of cancer, such as liver cell carcinoma, the elastin content is a major factor. The elasticity of a tumor depends on the ECM and the stiffness and is responsible for tissue homeostasis [35]. In colorectal cancer (CRC), elastin gene expression was recently examined and it was found that elastin decisively regulates tumor development and the microenvironment [36]. In this study, elastin gene expression was compared in CRC tumors from patients with adjacent non-tumorous colon tissue and healthy tissue (control). Elastin gene expression was found to be increased in patients with CRC tumors compared to the control group and adjacent non-tumor colon tissue. Metallopeptidase (MMP) 9 and 12 and TIMP3 were increased in the colon cancer cells. Another example is breast cancer, where elastin promotes the invasiveness of breast cancer cells [37].

The interaction between the tumor cells and the matrix protein elastin is mediated by elastin-binding proteins (EBPs), S-Gal, and Galectin-3 through the expression and release of elastases [23]. Comparing our two groups, 500 mm^3^ and 1000 mm^3^, showed that the group with smaller tumor volumes had a higher SI using the elastin-specific contrast agents. It can be concluded that smaller tumors can be detected particularly well due to a clear distinction from surrounding tissue. In contrast, tumors with a larger volume have fewer elastic fibers, which could be a clear signal of metastasis [23].

The expression level of elastin was not reflected in the Western blot (Figure 4D), as expected from the MRI images. In the Western blot we detected weak bands expressing elastin. The weak bands can be explained by the fact that elastin fibers are insoluble. Elastin is a cross-linked polymer whose cross-linking is difficult to break. Before cross-linking, the soluble precursor tropoelastin forms self-associated aggregates (coacervation) after expression [15]. Only these non-cross-linked aggregates can be broken down back into the small soluble tropoelastin proteins that can be detected in the Western blot. Thus, the Western blot showed the expression of tropoelastin and the coacervated elastin. The band intensity showed a higher expression of new elastin in the 500 mm^3^ compared to the 1000 mm^3^ tumors. This correlates to the higher amount of elastin in the immunofluorescence staining (Figure 2E) and the higher MRI signal (Figure 2A) in the 500 mm^3^ tumors. Since both MRIs showed a high increase in contrast in the T1 measurement after applying the elastin-specific probe, a high elastin density, especially in the periphery of the probe, could be estimated. The immunofluorescence staining against elastin as well as the EvG staining (Figure 4A,B) supports this thesis.

Through cell–matrix interaction, the extracellular matrix is constantly remodeled. The remodeling of the ECM creates a new microenvironment that promotes tumorigenesis and metastasis [38]. Elastin-derived matrikines promote tumor progression (for example, Val-Gly-Val-Ala-Pro-Gly or Ala-Gly-Val-Pro-Gly-Leu-Gly-Val-Gly) [38]. The degradation of elastin produces various proteolytic enzymes, elastases, and MMPs. Matrikins are able to activate the expression of MMPs, which positively promotes the tumor [38]. Elastin can help to detect a tumor or metastases at an early stage by morphologically changing the tumor and initiate appropriate therapy [23]. In addition, a therapy that specifically targets elastin peptides would be a possibility to reduce tumor growth and invasion.

Molecular imaging provides precise information about the tumor but also about the structural characteristics of the tumor. An important step is the use of molecular imaging techniques to make predictions about the molecular characteristics of the tumor to prevent invasive surgery. Currently, molecular imaging methods are based on cell metabolism, hormone receptors, and membrane proteins [39]. The cell metabolism of tumor cells differs from surrounding healthy cells, which can be exploited in molecular imaging. Current research is being conducted on radiolabeled analogs of the metabolic substrates choline, acetate, glucose, amino acids, and nucleosides [39]. These are not specific to the detection of malignant diseases. Specific imaging for PCa can also be achieved using androgen receptors and membrane proteins. For the development of such biomarkers, it is important that they be of low molecular weight and can therefore be released faster in the blood. Pu et al. (2016) showed the targeting of prostate-specific membrane antigen (PSMA) with a protein MRI contrast agent (ProCA) [40]. The 100 kDa glutamate carboxypeptide PSMA is involved in signal transduction, receptor function, nutrient uptake, and cell migration. It is overexpressed in epithelial cells of prostate cancer. PSMA can be detected in primary, secondary, and metastatic prostate cancer, which makes it a good marker [41]. Pu et al. demonstrated that the targeted MRI contrast agent has good Gd^3+^ binding affinity, metal selectivity, and relaxivity, and strong PSMA targeting ability [40]. The contrast agent (ProCA32.PSMA) showed a signal change in T1-weighted images in tumor-bearing mice (xenograft model), but also in the T2-weighted images [40]. The experiments were performed on a 7 Tesla MRI. The results are promising and can be implemented for early detection, but still need to be tested in an orthotopic model first.

Our results show the detection of tumors with components of the ECM in a clinical MRI, which can generate statements about tumor volume and enable predictions about the further course of a tumor. With the help of molecular imaging methods, it is possible to make individual disease predictions without taking tissue samples from the organism. Molecular imaging would be a good addition to existing commercial diagnostic possibilities. The main advantage of molecular MRI is to generate a non-invasive assessment at cellular level. An elastin-specific contrast agent has not only shown good results in cardiovascular diseases [42,43,44] but could also be used for the detection of malignant liver tumors. In a recent study, an elastin-specific contrast agent was used to visualize VX2-hepatic tumors in a rabbit model, and the use of the molecular agent to differentiate specific tumor and peritumoral regions based on its ECM composition was confirmed [31]. In addition, Sun et al. were able to demonstrate the usefulness of the probe even in kidney fibrosis [45].

A combination of available diagnostic techniques and molecular imaging would allow specific statements about the stage of disease in a non-invasive manner. If therapy is initiated at an early stage, the chances of survival for the affected patient will increase.

### Limitations

This study was conducted in a xenograft mouse model. This allows the tumor to grow in the organism outside the organ. An orthotopic mouse model would allow the tumor to grow in its natural microenvironment.

## 5. Conclusions

Our study demonstrates that molecular imaging using an elastin-specific gadolinium-containing contrast agent is feasible in prostate cancer. The study also confirms an apparent loss of elastin-specific ECM components in larger tumors. Such an imaging approach could be useful, for example, in predicting the metastatic potential of the tumor.

## Figures and Tables

**Figure 1 biology-10-01217-f001:**
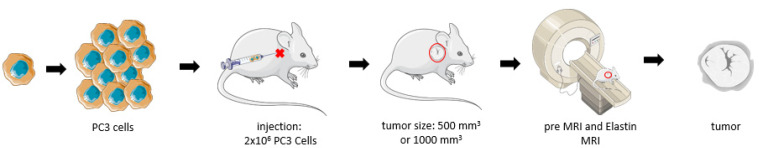
Overview of the experiment. Human PC3 cells were cultured in a cell-culture flask. A total of 2 × 10^6^ cells were subcutaneously injected into male SCID mice. Two different tumor sizes were achieved: 500 mm^3^ and 1000 mm^3^. After obtaining the desired tumor size, MR imaging was performed using an elastin-specific contrast agent. Tumor tissue was excised for ex vivo analysis.

**Figure 2 biology-10-01217-f002:**
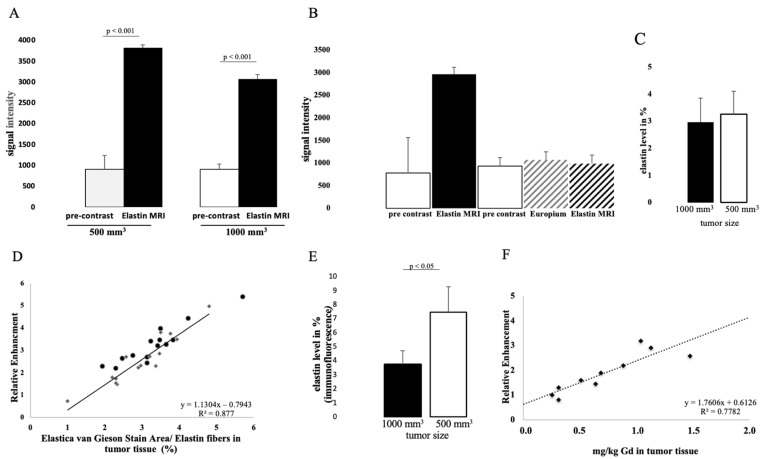
In vivo molecular MR imaging evaluation and quantification of prostate cancer using an elastin-specific contrast agent. (**A**) MRI measurements show the evaluation of MRI images (T1-weighted sequence) before and after contrast agent administration in two tumor volumes (1000 mm^3^ and 500 mm^3^). A total of 14 animals were examined per group (n = 14). After the elastin-specific contrast agent was administered, the value increased to an RE of 3037 (1000 mm^3^) and 3819 (500 mm^3^). The data are significant (*p* ≤ 0.001). (**B**) A competition experiment was performed to show the specific binding of the elastin-specific contrast agent. Three mice were used for this experiment (n = 3). On day 1, images were taken before and after the elastin-specific contrast agent administration. After 24 h (day 2) the animals were examined again. First a pre-contrast image was taken, then an elastin-specific probe with europium was administered (instead of Gd^3+^, it was conjugated with europium), and finally the elastin-specific contrast agent was applied. There was very little to no signal change. The data therefore show specific binding of the elastin-specific contrast agent. (**C**) Elastin levels of n = 14 animals per group were analyzed by histology. (**D**) The dot plot shows the correlation between MRI data (relative enhancement) and histological data. The Elastica–van Gieson stain was used to stain elastin fibers in the tumor tissue. The R-squared value is 0.88. (**E**) Elastin levels of n = 4 animals per group were analyzed by immunofluorescence. The data are significant (*p* ≤ 0.05). (**F**) shows the correlation between MRI data and Gd content in tumor tissue measured with ICP-MS. The R-squared value is 0.78.

**Figure 3 biology-10-01217-f003:**
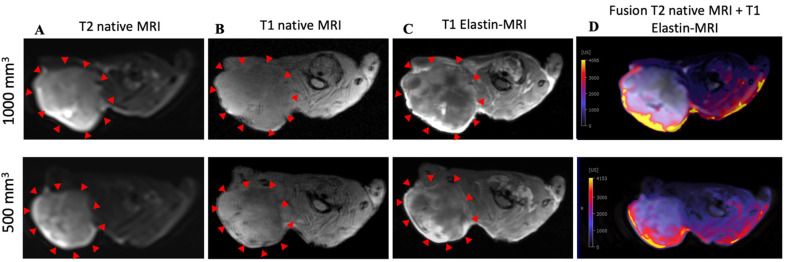
MRI and histological images of prostate cancer: histological characterization of elastin distribution in 1000 mm^3^ and 500 mm^3^ PC3 tumors. (**A**) shows a representative native MRI image of a T2-weighted sequence from the scapula area of a mouse that developed a tumor with a volume of 1000 mm ^3^ (top) and 500 mm^3^ (bottom). The red arrows indicate the tumor. (**B**) shows a representative native MRI image of a T1-weighted sequence from the scapula area of a SCID mouse that developed a tumor volume of 1000 mm^3^ (top) and 500 mm^3^ (bottom). (**C**) shows an MRI image of a T1-weighted sequence with administration of the elastin-specific contrast agent. A signal change in the tumor area after contrast agent administration is visible. The red arrows mark the total area of the tumor. In this region there are clear white/bright areas showing the signal change from the previous image (1000 mm^3^ (top) and 500 mm^3^ (bottom)). (**D**) shows a fusion of a native T2-weighted sequence and a T1-weighted sequence after administration of the elastin-specific contrast agent in the same mouse (1000 mm^3^ (top) and 500 mm^3^ (bottom)).

**Figure 4 biology-10-01217-f004:**
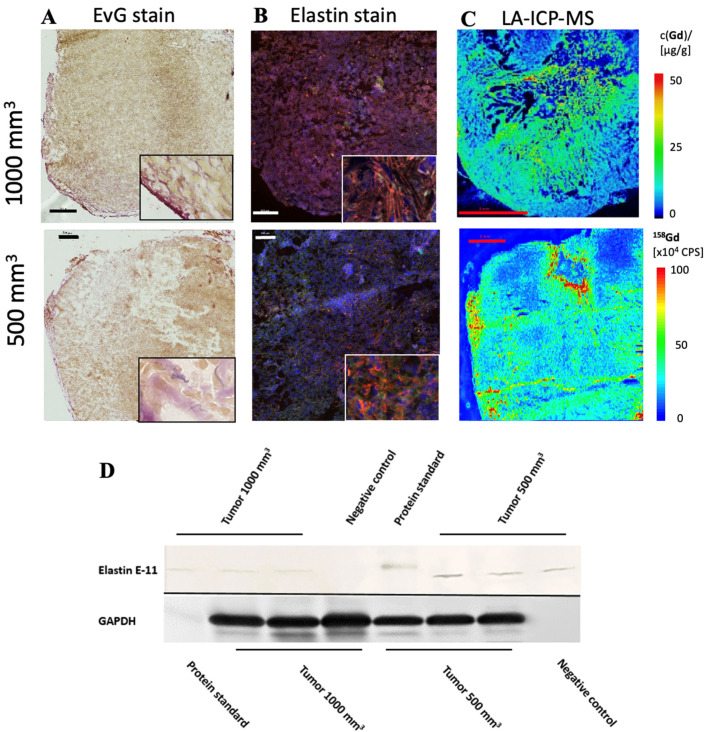
Histological characterization of elastin distribution in 1000 mm^3^ and 500 mm^3^ PC3 tumors. (**A**) shows Elastica–van Gieson staining in frozen sections with a slice thickness of 10 um from 1000 mm^3^ (top) and 500 mm^3^ (bottom) PC3 tumors. Elastic fibers are blue-violet. Elastin fibers were detected especially in the marginal area. In the lower right corner is an enlarged section of the image of the peripheral area of the tumor. Scale bar = 500 μm. (**B**) A parallel section of the same tumor (thickness 10 μm) as in A was prepared with an anti-elastin antibody (specific for immunofluorescence). The elastic fibers are visible in red. Staining of the cell nuclei was achieved using Dapi (blue). Scale bar = 500 μm; 1000 mm^3^ (top) and 500 mm^3^ (bottom) PC3 tumors. (**C**) The element gadolinium was detected by LA-ICP-MS. The scale shows the intensity of the detected gadolinium (cps) (red—high to blue—low). Scale bar = 2 mm; 1000 mm^3^ (top) and 500 mm^3^ (bottom) PC3 tumors. (**D**) For each group, 3 tumors (n = 3 per group) were used for Western blot analysis to detect the expression of elastin E-11. Here, a different antibody was used than for immunofluorescence, as the antibody is specific for Western blot analyses. GAPDH was included to control protein levels.

**Table 1 biology-10-01217-t001:** Experimental parameters of iCAP Qc.

Parameter	Value
Power (W)	1550
Nebulizer gas flow rate (L min^−1^)	1.08
Aux gas flow rate (L min^−1^)	0.65
Cool gas flow rate (L min^−1^)	14
Sample flow rate (mL min^−1^)	0.40
Dwell time [ms]	0.01
Isotopes monitored	^103^Rh, ^155^Gd, ^156^Gd, ^157^Gd, ^158^Gd, ^160^Gd,

## Data Availability

Data are available from the corresponding author upon reasonable request.

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
