# Peer review of "Visualization and Quantification of the Extracellular Matrix in Prostate Cancer Using an Elastin Specific Molecular Probe"

_biology, 2021, doi:10.3390/biology10111217_

Round 1

Reviewer 1 Report

This is a study on prostate cancer in a xenograft mouse model, evaluating the extracellular matrix using an elastin-specific molecular probe for molecular MRI. Based on the results, the authors concluded that molecular MRI could become a novel, quantitative tool for the early evaluation and detection of prostate cancer.

The paper is very well written with a strong methodology. Herein are my comments:

  1. Introduction: the authors nicely presented the limitations of PSA in the diagnosis of prostate cancer. However, the quote from Izumi et al.’s paper (lines 67 and 68) is not correct and should be modified.
  2. mpMRI has been gaining acceptance in the diagnosis of prostate cancer in the years, with high sensitivity and specificity. What are the advantages of molecular MRI over mpMRI? The authors should elaborate on this to justify their research question.
  3. Is there any data available regarding the association of the molecular MRI findings and mpMRI results (i.e., PIRADS) and path grading (i.e., Gleason)?
  4. This study mainly confirmed the feasibility (not reliability or specificity) of the molecular MRI in detecting prostate cancer. The conclusion should be edited.

Author Response

Reviewer 1

1.) Introduction: the authors nicely presented the limitations of PSA in the diagnosis of prostate cancer. However, the quote from Izumi et al.’s paper (lines 67 and 68) is not correct and should be modified.

We are thankful for this comment, we modified the according part.

Line 67-70: “Due to its relatively low specificity and a low sensitivity, it leads to many false positive diagnoses [1, 2]. As a result, many patients undergo unnecessary prostate biopsy [2]. In patients with a normal PSA level, PCa could be diagnosed in 30% of cases, with 10% being assigned to aggressive PCa [1].”

2.) mpMRI has been gaining acceptance in the diagnosis of prostate cancer in the years, with high sensitivity and specificity. What are the advantages of molecular MRI over mpMRI? The authors should elaborate on this to justify their research question.

We are thankful for this comment. We agree that mpMRI is specific and also sensitive, but often the results are interpreted differently by clinicians. In addition, a tissue biopsy is still needed, which is an invasive procedure. Therefore, a non-invasive technique is needed. Here, molecular imaging could help or be applied in a supporting method to the current methods. I have included the problem in the introduction (lines 84-89): mpMRI has a high sensitivity and specificity, but it has a low positive predictive value (PPV) [3]. The Prostate Imaging Reporting and Data System (PI-RADS) helps to detect PCa in a standardized form. The main challenge is the different interpretation of the PI-RADS results by clinicians and medical staff [3]. PCa mpMRI diagnosis must finally be verified by biopsy. Therefore, it is necessary to develop a non-invasive screening method with high specificity. “

3.) Is there any data available regarding the association of the molecular MRI findings and mpMRI results (i.e., PIRADS) and path grading (i.e., Gleason)?

We thank the reviewer for this important question. In the current study we specifically focused on the molecular MRI findings and did to acquire all sequences required for a dedicated PIRADs analysis. We are however planning a specific PIRADS and Gleason analysis for the follow up study to analyse the additional value of molecular MRI in this context.

4.) This study mainly confirmed the feasibility (not reliability or specificity) of the molecular MRI in detecting prostate cancer. The conclusion should be edited.

We thank the reviewer for his comment. We have adjusted the Conclusion.

Line: 478- 482: “Our study demonstrated that molecular imaging using an elastin-specific gadolinium-containing contrast agent is feasible in prostate cancer. The study also confirms an apparent loss of elastin-specific ECM components in larger tumors. Such an imaging approach could be useful, for example, in predicting the metastatic potential of the tumor.”

  1. Kim, J.H. and S.K. Hong, Clinical utility of current biomarkers for prostate cancer detection. Investig Clin Urol, 2021. 62(1): p. 1-13.
  2. Mistry, K. and G. Cable, Meta-Analysis of Prostate-Specific Antigen and Digital Rectal Examination as Screening Tests for Prostate Carcinoma. The Journal of the American Board of Family Practice, 2003. 16: p. 95-101.
  3. Westphalen, A.C., et al., Variability of the Positive Predictive Value of PI-RADS for Prostate MRI across 26 Centers: Experience of the Society of Abdominal Radiology Prostate Cancer Disease-focused Panel.Radiology, 2020. 296(1): p. 76-84.

Reviewer 2 Report

This work is devoted to the study of the expression of the extracellular matrix protein elastin in the development of diagnostics of prostate cancer. This search is important because the currently known methods do not always provide a reliable estimate of cancer. The study assesses the level of elastin in two tumors of visually different sizes after approximately the same period, from 30 to 64 days. The results obtained suggest that the use of a gadolinium-based marker for elastin holds promise for evaluating tumor size increase as an indicator of cancer progression in a xenographic mouse model. 

My main criticism stems from the observation of Western blotting that elastin is a poorly expressed protein that is almost invisible in 1000 mm3 samples and very faintly visible in 500 mm3 samples when detected with anti-elastin mAb. This casts doubt on the choice of elastin as a promising marker in the suggested diagnosis in the Conclusion. Hence, this requires an explanation.

In the same regard, it is not clear how correlation between MRI data and histology is important for diagnosis, if elastin is clearly more expressed in the figure 3D than in 4D, and in figures 3F and 4F this is difficult to assess (no data) the immunofluorescent response.

Small remarks.

Why were different anti-elastin antibodies used in histological and Western blot analyzes?

line 84: State here what Gd is.

It seems to me that Figures 3 and 4 can be combined into one using upper- and lower-case letters. Indeed, the legends are the same, except for 1000 and 500 mm3. This change will make it easier for the reader to compare the results.

Lines 443-446: It's not clear how this relates to your data. You have not shown the effect of stress and elastin degradation.

Author Response

  1. My main criticism stems from the observation of Western blotting that elastin is a poorly expressed protein that is almost invisible in 1000 mm3samples and very faintly visible in 500 mm3 samples when detected with anti-elastin mAb. This casts doubt on the choice of elastin as a promising marker in the suggested diagnosis in the Conclusion. Hence, this requires an explanation.

We are thankful for the request. We agree that the lines are very weak. We believe that the challenge is not the poor expression or the small amount of elastin in the ECM of the tumor, but the insolubility of elastin [3]. Since elastin is a crosslinked polymer, even the RIPA buffer is not able to break the crosslinking. Before the crosslinking, the soluble precursor tropoelastin forms rapidly after expression self-associated aggregates (coacervation) [1]. Only these non-crosslinked aggregates can be broken into the smaller, soluble proteins, which we detect in the WB. The weak lines in the WB shows a higher expression of new elastin in the 500 mm3 compared to the 1000 mm3, which can be linked to the elastin density in the probe.

  1. In the same regard, it is not clear how correlation between MRI data and histology is important for diagnosis, if elastin is clearly more expressed in the figure 3D than in 4D, and in figures 3F and 4F this is difficult to assess (no data) the immunofluorescent response.

We thank the reviewer for his comments. The fusion of T2 native MRI and T1 Elastin-MRI figures (3D and 4D) does not specifically reflect the elastin expression, since the contrast in the 1000 mm3 tumor is already higher than in the 500 mm3 tumor and the MRI gives no statement about protein expression. We made this now clearer in the figure legend. We could demonstrate by comparison the native T1 MRI and the T1 MRI after ESMA administration, a higher Signal intensity (SI) for the 500 mm3 compared to the 1000 mm3, like the diagram in Figure 2A showed. Additionally, we could link an increased contrast to an increasing signal in the EvG staining.  Fusion imaging only reflects the relative signal intensity between the T2 and T1 sequences in the same mouse. We stated this now clearer in the figure legend to avoid any confusion for the reader.

We analyzed the immunofluorescent data and included the data in the manuscript.
Figure 2E and Lines 352-355: To further evaluate the distribution of elastic fibers in the PC3 tumor, an immunofluorescence staining with an anti-elastin antibody was performed (Figure 4B). This showed an irregular distribution of the elastin fibers in the tissue. The mean value of n=4 was 7.5% (σ = 1.8) in 500 mm3 and 3.7% (σ = 0.9) in 1000 mm3 tumors (p < 0.05).”

Small remarks.

  1. Why were different anti-elastin antibodies used in histological and Western blot analyzes?

We are thankful for the question of the reviewer. For the histological part, we used the antibody from abcamÒ, which is especially suited for the use in histological analysis. Nevertheless, this antibody is not suited for Western Blot analyzes (please see the data sheet). Therefore, we had to change for the Western Blot analysis the antibody. The antibody from Santa CruzÒ is well suited for WB. Additionally, Santa CruzÒ provides a positive control sample for Western Blot especially for this Antibody. We are now specifically explaining this in the methods section.

  1. line 84: State here what Gd is.

We are grateful for this comment. We will change the according paragraph in the manuscript to make it more clear.
Line 90-96: “For improvement of MRI contrast, the paramagnetic lanthanide metalion gadolinium (III) (Gd3+) can be used in a complex with organic chelates, like the macrocyclic DOTA (2,2′,2′′,2′′′-(1,4,7,10-Tetraazacyclododecane-1,4,7,10-tetrayl)tetraacetic acid) or the linear DTPA (2,2′,2′′,2′′′-{[(Carboxymethyl)azanediyl]bis(ethane-2,1-diylnitrilo)}tetraacetic acid) as contrast media. Thereby, the unpaired electrons in the Gd-ion will shorten the T1 relaxivity of the neighboring water protons and, thus, the signal intensity of the Ta-weighed image will be increased [2]. “

  1. It seems to me that Figures 3 and 4 can be combined into one using upper- and lower-case letters. Indeed, the legends are the same, except for 1000 and 500 mm3. This change will make it easier for the reader to compare the results.

We thank the reviewer for his suggestion for improvement. In the revised version we combined the MRI figures in Figure 3 and the histology figures in Figure 4. The comparison between the 1000 and the 500 mm3 tumors makes this clearer for the reader. Nevertheless, we found one figure to potentially be too complex for the reader, that’s why we decided to separate them by the in vivo and ex vivo examination.

  1. Lines 443-446: It's not clear how this relates to your data. You have not shown the effect of stress and elastin degradation.

Thank you for the comment and have removed this part in the discussion.

  1. Wise, S.G. and A.S. Weiss, Tropoelastin. Int J Biochem Cell Biol, 2009. 41(3): p. 494-7.
  2. Xiao, Y.D., et al., MRI contrast agents: Classification and application (Review). Int J Mol Med, 2016. 38(5): p. 1319-1326.
  3. Yeo, G. C., Keeley, F. W., & Weiss, A. S. (2011). Coacervation oftropoelastin. Advances in
    colloid and interface science
    167(1-2), 94-103.

Round 2

Reviewer 2 Report

The authors took into account the remarks of the reviewer and the manuscript has been significantly improved. 1. However, the remark about the low level of elastin expression was ignored. Typically, a candidate diagnostic marker should be well expressed in cancer cells. Please explain this discrepancy as your work is focused on the possible use of elastin to more accurately diagnose prostate cancer. 2. Indicate the presence of two elastin proteins of different sizes, which can be assumed for the two tumors in Figure 4. The authors explain to the reviewer the possible reason for the low expression due to the insolubility of elastin in tumors of different sizes. I think this explanation should be included for the readers in the Discussion. To some extent, this reflects my first remark. 3. Lack of references to Fig. 3 B, C, D in the text.

Author Response

1- However, the remark about the low level of elastin expression was ignored. Typically, a candidate diagnostic marker should be well expressed in cancer cells. Please explain this discrepancy as your work is focused on the possible use of elastin to more accurately diagnose prostate cancer. 2- Indicate the presence of two elastin proteins of different sizes, which can be assumed for the two tumors in Figure 4. The authors explain to the reviewer the possible reason for the low expression due to the insolubility of elastin in tumors of different sizes. I think this explanation should be included for the readers in the Discussion. To some extent, this reflects my first remark.

We thank the reviewer for the comments and apologize for not being more clear. We agree that candidate diagnostic markers should be particularly well expressed in the target and are convinced that elastin fulfills this criterion, as a highly expressed extracellular matrix protein. The low level detection in the western blots is associated with the highly insoluble character of elastin as a crosslinked polymer, which is not broken apart by western blot loading buffer.

The soluble part is the precursor of elastin, the tropoelastin, which coacervate before crosslinking. Only the coacervated elastin and the free tropoelastin can be measured by western blot and thus, the western blot can only be used to compare the expression level of tropoelastin since the antibody is also staining tropoelastin. The total amount of elastin can therefore only be determined indirectly with this technique. Nevertheless, the use of our elastin specific probe, ESMA, which showed high selectivity against elastin and low unspecific binding, lead to a high increase of the MRI T1 sequence, like we showed in picture 2a, indicates a high elastin level in the tumor probes. Additionally, the very sensitive immunofluorescence against elastin showed a significant higher elastin level in the 500 mm3 tumor compared to the 1000 mm3 tumor. These findings correlate with the higher tropoelastin expression and the higher MRI contrast in the 500 mm3 tumor.  

All these results suggest that elastin can be a good marker for tumor disease and that an elastin-specific contrast agent can be used in the early diagnosis of prostate cancer in addition to the current methods.

We added according passages to the text to make this clearer for the reader.

Lines 429-443
“The expression level of elastin was not reflected in the Western blot (Figure 4D) as expected from the MRI images. In the western blot we detected weak bands expressing elastin. The weak bands can be explained by the fact that elastin fibres are insoluble. Elastin is a cross-linked polymer that is difficult to break this crosslinking. Before cross-linking, the soluble precursor tropoelastin forms self-associated aggregates (coacervation) after expression [15]. Only these non-cross-linked aggregates can be broken down back into the small soluble tropoelastin proteins that can be detected in the Western blot. Thus, the western blot shows the expression of tropoelastin and the coacervated elastin. The band intensity shows a higher expression of new elastin in the 500 mm3 compared to the 1000 mm3 tumors. These correlates to the higher amount of elastin in the immunofluorescence staining (Figure 2E) and the higher MRI signal (Figure 2a) in the 500 mm3tumors. Since both MRIs show a high increase of contrast in the T1 measurement after applying the elastin specific probe, a high elastin density especially in the periphery of the probe can be estimated. The immunofluorescence staining against elastin as well as the EvG staining (Figure 4A&B) supports this thesis.”

We also make reference to the different antibodies in the manuscript.
Lines: 360-362
“The antibody used for the western blot is different from that used for immunofluorescence, as the respective antibodies had to be applied specifically to one method.”

Figure 4 legend:

“(B) A parallel section of the same tumor (thickness 10 mm) as in A was prepared with an anti-elastin antibody (specific for immunofluorescence). The elastic fibers are visible in red. Staining of the cell nuclei was achieved using Dapi (blue). Scale bar = 500 mm; 1000 mm3 (top) and 500 mm3 (bottom)-PC3 tumor.

(D) For each group, 3 tumors (n = 3 per group) were used for western blot analysis to detect the expression of elastin E-11. Here, a different antibody was used than for immunofluorescence, as the antibody is specific for Western blot analyses. GAPDH was included to control protein levels.”

3. Lack of references to Fig. 3 B, C, D in the text.

We thank the reviewer for this comment. We have added parts in the manuscript.

Lines: 308-309

“Figure 3B shows a pre-contrast image and Figure 3C shows an image with contrast medium. A good difference between the two groups can already be seen here.”

 Lines: 323-325

“To show signal enhancement within a mouse, a fusion map was created (Figure 3D) which shows the SI between the T2 and T1 sequences after contrast administration in the same mouse.”
